# Tracking Control of a Hyperchaotic Complex System and Its Fractional-Order Generalization

Feng Liang [1,2], Lu Lu [2,†], Zhengfeng Li [3], Fangfang Zhang [3,*] and Shuaihu Zhang [3]

1   Department of Computer and Software Engineering, Shandong College of Electronic Technology, Jinan 250200, China; lf1839@163.com
2   College of Artificial Intelligence and Big Data for Medical Sciences, Shandong First Medical University, Shandong Academy of Medical Sciences, Jinan 250117, China; squarelu@hotmail.com
3   School of Information and Automation Engineering, Qilu University of Technology (Shandong Academy of Sciences), Jinan 250353, China; zhengf_li@163.com (Z.L.); 18437902858@163.com (S.Z.)
*   Correspondence: zhff4u@qlu.edu.cn; Tel.: +86-151-6916-3922
†   These authors contributed equally to this work.

**Abstract:** Hyperchaotic complex behaviors often occur in nature. Some chaotic behaviors are harmful, while others are beneficial. As for harmful behaviors, we hope to transform them into expected behaviors. For beneficial behaviors, we want to enhance their chaotic characteristics. Aiming at the harmful hyperchaotic complex system, a tracking controller was designed to produce the hyperchaotic complex system track common expectation system. We selected sine function, constant, and complex Lorenz chaotic system as target systems and verified the effectiveness by mathematical proof and simulation experiments. Aiming at the beneficial hyperchaotic complex phenomenon, this paper extended the hyperchaotic complex system to the fractional order because the fractional order has more complex dynamic characteristics. The influences order change and parameter change on the evolution process of the system were analyzed and observed by MATLAB simulation.

**Keywords:** hyperchaotic complex system; tracking control; fractional order

## 1. Introduction

Chaos is a complex nonlinear phenomenon in nature. The chaotic system has extensive applications in the fields of secure communication [1], industrial process [2], ecosystem [3], and so on. With the application of chaos theory in increasingly more fields, there are increasingly more requirements for chaotic systems. For example, people expect higher dimensional and more complex chaotic systems to describe industrial processes. Therefore, following this, scholars put forward the hyperchaotic system, complex chaotic system, and hyperchaotic complex system [4–6]. The hyperchaotic complex system in particular has higher dimensions and more controllable parameters [7,8] that can more accurately describe some chaotic phenomena in the industrial process. The chaotic system will experience four states: stable point, period, chaos, and divergence. When the system is in the periodic or period doubling state, the phase plane will form a closed trajectory, that is, the limit cycle. Generally speaking, the chaotic system will evolve from a periodic state to a chaotic state, and the corresponding phase diagram will evolve from limit cycle to chaotic attractor [9].

However, some chaotic behaviors are harmful, while some are beneficial. As for harmful behaviors, we hope to transform them into expected behaviors. For example, permanent magnet motor under some parameters can produce chaotic behaviors and disturb the normal operation of the motor. Therefore, it is necessary to add a controller to make it track the desired motion trajectory [10,11]. This means the tracking control for chaos, which can be used to obtain the desired output and improve the performance of the system [12,13]. The research on the tracking control of hyperchaotic complex systems is of great value.

At present, most of the research focuses on the tracking control of real chaotic systems [14–17], and few scholars have studied the tracking control of hyperchaotic complex systems. Gao proposed a novel tracking control method for Lorenz systems by using single-state feedback [18]. Loria addressed the problem of controlled synchronization of a class of uncertain chaotic systems [19]. Zhang presented the tracking control method and the parameter identification procedure, aiming at CVCSs with complex parameters [20]. Chaudhary et al. investigated a hybrid projective combination–combination synchronization scheme (HPCCSS) in four different hyperchaotic (HC) systems via active control technique (ACT) [21]. Abbasi proposed a robust resilient design methodology for stabilization and tracking control for a class of chaotic dynamical systems [22]. Zhao realized tracking control and synchronization of the fractional hyper chaotic Lorenz system [23]. Nagy et al. investigated the combination synchronization phenomena of various fractional-order systems using the scaling matrix [24]. In this paper, the tracking control of complex hyperchaotic system was realized.

As for beneficial chaotic behavior, we hope to enhance its chaotic characteristics. For example, more complex chaotic behaviors can make the stirring more sufficient in industrial process. Moreover, in secure communication, complex chaotic signal can increase the confidentiality of transmitted signal.

In order to obtain more complicated chaotic behaviors, we can extend the integer-order hyperchaotic complex systems to fractional order, because fractional order can increase the degree of freedom of the chaotic system and make its dynamic behaviors more complicated. As an extension of the integer-order complex hyperchaos system, the fractional- order complex hyperchaos system also has higher accuracy in describing processes in many fields and can more accurately describe various irregular physical phenomena. Therefore, many scholars have carried out a large amount of research on the fractional order. Ma investigated a new 4D incommensurate fractional-order chaotic system [25]. Jahanshahi investigated a multi-stable fractional-order chaotic system [26]. Rahman presented a new three-dimensional fractional-order complex chaotic system [27].

However, the above references fractional-order extension of real chaotic systems, but few scholars have studied the fractional-order hyperchaotic complex systems. In this paper, the complex hyperchaotic system was extended to fractional order.

On the basis of the above discussion, the main innovations of this paper are as follows:

(1) The tracking controller for the hyperchaotic complex system is designed. Three state variables of the hyperchaotic complex system track the sine function, constant, and complex Lorenz chaotic system individually, which realize the control of harmful chaotic behaviors. The stability and feasibility of the controller were verified by mathematical proof and simulation experiments.

(2) The hyperchaotic complex system is extended to fractional order, which enhances the beneficial chaotic behaviors. The effects of initial value, order, and parameters on the fractional hyperchaotic complex system are discussed.

The rest of this paper is structured as follows: In Section 2, we introduce the model of hyperchaotic complex system. In Section 3, we designed a controller to realize the tracking control of the hyperchaotic complex system and verified its feasibility from two aspects of a mathematical proof and simulation experiment. In Section 4, we extended the complex hyperchaotic system to fractional order and analyzed its initial value sensitivity and the system evolution process of the fractional complex hyperchaotic system with order and parameters. In the last section, we conclude the paper.

## 2. The Model of the Hyperchaotic Complex System

In 2021, Li et al. constructed a new hyperchaotic complex system by adding feedback term and introducing complex variables [7]. The mathematical model is as follows:

$$\begin{cases} \dot{x}_1 = a(x_2 - x_1) + x_2 x_3 + x_4 \\ \dot{x}_2 = cx_1 - x_2 - x_1 x_3 + x_4 \\ \dot{x}_3 = 1/2(\overline{x}_1 x_2 + x_1 \overline{x}_2) - bx_3 \\ \dot{x}_4 = 1/2(\overline{x}_1 x_2 + x_1 \overline{x}_2) - dx_4 \end{cases} \tag{1}$$

where $\dot{x}_1 = u_1 + ju_2, \dot{x}_2 = u_3 + ju_4$ is a complex variable and $\dot{x}_3 = u_5, \dot{x}_4 = u_6$ is a real variable. We separated the real part and imaginary part of the variable to obtain the equivalent mathematical model, as shown in system (2):

$$\begin{cases} \dot{u}_1 = a(u_3 - u_1) + u_3 u_5 + u_6 \\ \dot{u}_2 = a(u_4 - u_2) + u_4 u_5 \\ \dot{u}_3 = cu_2 - u_3 - u_1 u_5 + u_6 \\ \dot{u}_4 = cu_2 - u_4 - u_2 u_5 \\ \dot{u}_5 = u_1 u_3 + u_2 u_4 - bu_5 \\ \dot{u}_6 = u_1 u_3 + u_2 u_4 - du_6 \end{cases} \tag{2}$$

When $a = 10, b = 8/3, c = 20, d = 15$ and the initial value is (1, 2, 3, 4, 5, 6), the system shows obvious chaotic characteristics. The Lyapunov exponents of system (2) is as follows: $LE_1 = 1.7555, LE_2 = 0.1175, LE_3 = 0, LE_4 = -11.6914, LE_5 = -14.8827, LE_6 = -22.3471$, which is $(+, +, 0, -, -, -)$, so the system is in a hyperchaotic state. The attractor phase diagram of system (2) is shown in Figure 1.

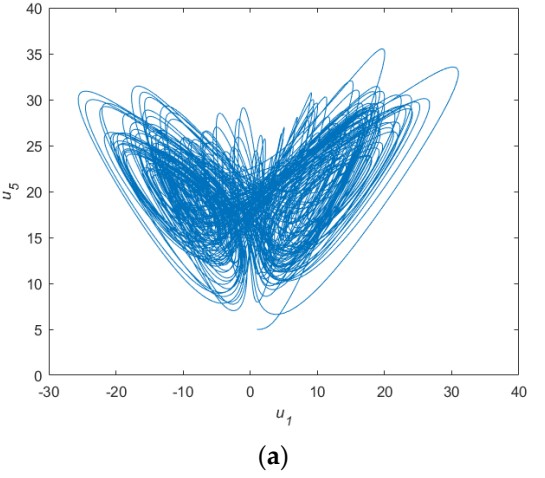
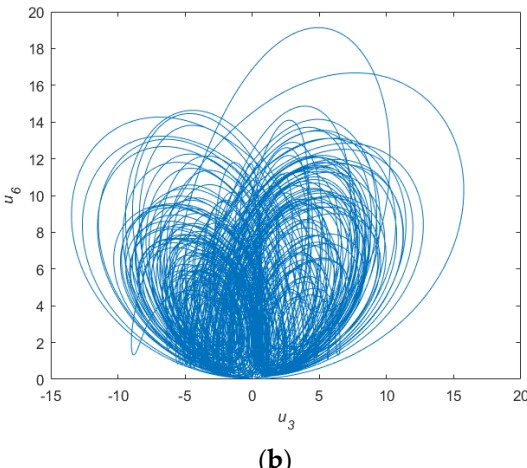

(**a**)        (**b**)

**Figure 1.** Attractor phase diagram of system (2). (**a**) $u_1 - u_5$ phase diagram of system (2); (**b**) $u_3 - u_6$ phase diagram of system (2).

## 3. Tracking Control

In this section, the controller is designed so that the three state variables of the system (2) can track the sine function, constant 5, and the fourth dimension of the complex Lorenz chaotic system individually.

The system model of complex Lorenz chaotic system is as follows:

$$\begin{cases} \dot{x}_1 = a_1(x_2 - x_1) \\ \dot{x}_2 = a_2 x_1 - x_1 x_3 - a_3 x_2 \\ \dot{x}_3 = -a_4 x_3 + 1/2(\overline{x}_1 x_2 - x_1 \overline{x}_2) \end{cases} \tag{3}$$

Separating the real part and imaginary part of the complex Lorenz chaotic system, the following equivalent mathematical model can be obtained:

$$\begin{cases} \dot{x}_1^r = a_1(x_2^r - x_1^r) \\ \dot{x}_1^i = a_1(x_2^i - x_1^i) \\ \dot{x}_2^r = a_2 x_1^r - x_1^r x_3 - a_3 x_2^r \\ \dot{x}_2^i = a_2 x_1^i - x_1^i x_3 - a_3 x_2^i \\ \dot{x}_3 = -a_4 x_3 + (x_1^r x_2^r + x_1^i x_2^i) \end{cases} \tag{4}$$

We add the controller to system (2) and obtain system (5)

$$\begin{cases} \dot{u}_1 = a(u_3 - u_1) + u_3 u_5 + u_6 + v_1 \\ \dot{u}_2 = a(u_4 - u_2) + u_4 u_5 + v_2 \\ \dot{u}_3 = cu_2 - u_3 - u_1 u_5 + u_6 + v_3 \\ \dot{u}_4 = cu_2 - u_4 - u_2 u_5 + v_4 \\ \dot{u}_5 = u_1 u_3 + u_2 u_4 - bu_5 + v_5 \\ \dot{u}_6 = u_1 u_3 + u_2 u_4 - du_6 + v_6 \end{cases} \tag{5}$$

where $V = (v_1, v_2, v_3, v_4, v_5, v_6)^T$ is the designed controller vector. We obtain the following Theorem 1.

**Theorem 1.** *As for system (5), if we design the controller as system (6),*

$$\begin{cases} v_1 = \dot{r}_1 + r_1 + (a-1)u_1 - au_3 - u_3 u_5 - u_6 \\ v_2 = \dot{r}_1 + r_1 + (a-1)u_2 - au_4 - u_4 u_5 \\ v_3 = \dot{r}_2 + r_2 - cu_2 + u_1 u_5 - u_6 \\ v_4 = \dot{r}_3 + r_3 - cu_2 + u_2 u_5 \\ v_5 = \dot{r}_2 + r_2 - u_1 u_3 - u_2 u_4 + (b-1)u_5 \\ v_6 = \dot{r}_2 + r_2 - u_1 u_3 - u_2 u_4 + (d-1)u_6 \end{cases} \tag{6}$$

*then the state variable $u_1$ can track sine function, the state variable $u_2$ can track constant 5, and the state variable can track $x_2^i$ of the complex Lorenz chaotic system.*

**Proof:** Set the error as $e_1, e_2, e_3$, then

$$e_1 = u_1 - r_1, e_2 = u_3 - r_2, e_3 = u_4 - r_3 \tag{7}$$

The expected goals are expressed as follows,

$$\begin{cases} r_1 = \sin t \\ r_2 = 5 \\ r_3 = x_2^i \end{cases} \Rightarrow \begin{cases} \dot{r}_1 = \cos t \\ \dot{r}_2 = 0 \\ \dot{r}_3 = a_2 x_1^i - a_3 x_2^i - x_1^i x_3 \end{cases} \tag{8}$$

Select Lyapunov function $V = \frac{1}{2}(e_1 + e_2 + e_3)^2 > 0$. Substituting (5)–(8) into Lyapunov function $V$, we can obtain

$$\begin{aligned} \dot{V} &= e_1 \dot{e}_1 + e_2 \dot{e}_2 + e_3 \dot{e}_3 \\ &= (u_1 - r_1)(u_1 - r_1)' + (u_3 - r_2)(u_3 - r_2)' + (u_4 - r_3)(u_4 - r_3)' \\ &= (u_1 - \sin t)(\sin t - u_1) + (u_3 - r_2)(5 - u_3) \\ &\quad + (u_4 - x_2^i)(cu_2 - u_4 - u_2 u_5 + v_4 - a_2 x_1^i + a_3 x_2^i + x_1^i x_3) \\ &= (u_1 - \sin t)(\sin t - u_1) + (u_3 - 5)(5 - u_3) + (u_4 - x_2^i)(x_2^i - u_4) \\ &= -(u_1 - \sin t)^2 - (u_3 - 5)^2 - (u_4 - x_2^i)^2 < 0 \end{aligned}$$

According to the Lyapunov stability theorem, the error of tracking control approaches 0, and the proof is completed. □

The tracking results are shown in Figures 2–4.

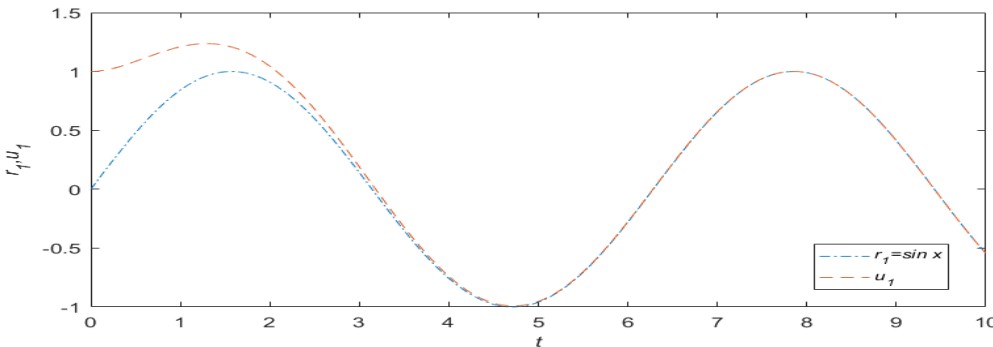

**Figure 2.** Tracking sine function.

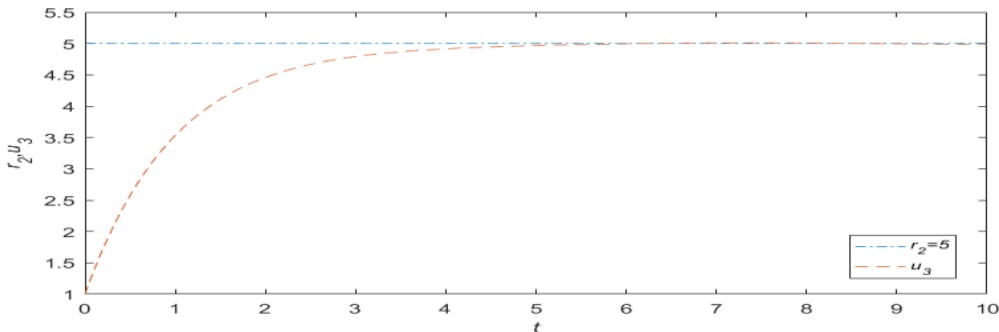

**Figure 3.** Tracking constant 5.

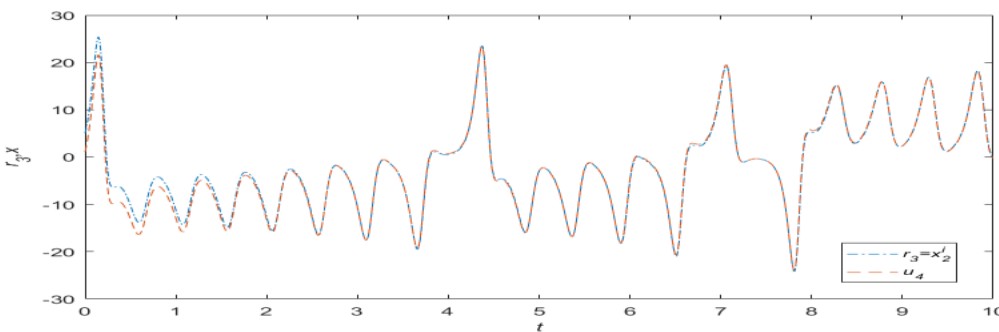

**Figure 4.** Tracking the fourth dimension of the complex Lorenz chaotic system.

The simulation results are consistent with the theoretical analysis, and the effectiveness of the controller is verified from two aspects of simulation and mathematical analysis.

## 4. Fractional-Order Generalization

In order to make the beneficial chaotic behavior in industrial process more complicated, this paper extended the hyperchaotic complex system to fractional order, which is a useful and simple method to enhance the beneficial chaotic behaviors.

### 4.1. Mathematical Background

Fractional order refers to any order of calculus. In a sense, fractional calculus is a generalized form of integer calculus. For fractional calculus, there are three main definitions: Grunwald Letnikov definition, Riemanu Liouville definition, and Caputo definition. Since Caputo definition includes initial conditions and initial values, Caputo calculus is considered in engineering calculations. In this paper, we chose the Caputo definition.

**Definition 1.** *Caputo fractional differential form is as follows*:

$$\,_0^C D_t^\alpha f(t) = \frac{d^m}{dt^m} J_{m-\alpha} = \begin{cases} \frac{1}{\Gamma(m-\alpha)} \int_0^t \frac{f(m)(\tau)}{(t-\tau)^{\alpha-m+1}} d\tau, & m-1 < \alpha < m \\ \frac{d^m}{dt^m} f(t), & \alpha = m \end{cases} \tag{9}$$

*where $\alpha = [m]+1$, $[m]$ is the integer part of $m$, $\Gamma(*)$ is the gamma function, and $D_t^\alpha$ is the gamma function $\alpha$ order differential operator. In this paper, $D_*^\alpha$ is used to represent $\,_0^C D_*^\alpha$, and we mainly consider the case of $0 < \alpha < 1$.*

### 4.2. Fractional-Order System Model

In this section, the hyperchaotic complex system is extended to fractional order, and the following fractional-order new hyperchaotic complex system is constructed.

$$\begin{cases} D_*^{\alpha_1} x_1 = a(x_2 - x_1) + x_2 x_3 + x_4 \\ D_*^{\alpha_2} x_2 = cx_1 - x_2 - x_1 x_3 + x_4 \\ D_*^{\alpha_3} x_3 = 1/2(\bar{x}_1 x_2 + x_1 \bar{x}_2) - bx_3 \\ D_*^{\alpha_4} x_4 = 1/2(\bar{x}_1 x_2 + x_1 \bar{x}_2) - dx_4 \end{cases} \tag{10}$$

where $D_*^{\alpha_l}$ is the Caputo operation of order $\alpha_l$, and $\alpha_l$ is the order of relevant variables of $x_l(l = 1, 2, 3, 4)$. According to fractional linear operation, we can obtain $D_*^{\alpha_1} x_1 = D_*^{\alpha_1}(u_1 + ju_2) = D_*^{\alpha_1} u_1 + jD_*^{\alpha_1} u_2, D_*^{\alpha_2} x_2 = D_*^{\alpha_2}(u_3 + ju_4) = D_*^{\alpha_2} u_3 + jD_*^{\alpha_2} u_4$ $D_*^{\alpha_3} x_3 = D_*^{\alpha_3} u_5, D_*^{\alpha_4} x_4 = D_*^{\alpha_4} u_6$.

The above system can be transformed into the following forms:

$$\begin{cases} D_*^{\alpha_1} u_1 = a(u_3 - u_1) + u_3 u_5 + u_6 \\ D_*^{\alpha_1} u_2 = a(u_4 - u_2) + u_4 u_5 \\ D_*^{\alpha_2} u_3 = cu_1 - u_3 - u_1 u_5 + u_6 \\ D_*^{\alpha_2} u_4 = cu_2 - u_4 - u_2 u_5 \\ D_*^{\alpha_3} u_5 = u_1 u_3 + u_2 u_4 - bu_5 \\ D_*^{\alpha_4} u_6 = u_1 u_3 + u_2 u_4 - du_6 \end{cases} \tag{11}$$

### 4.3. Fractional-Order Attractor

We select the initial value of system (11) as $(1, 1, 1, 1, 1, 1)$, $a = 10, b = 8/3, c = 30, d = 12$, and fractional-order $\alpha_l = 0.95(l = 1, 2, 3, 4)$. On the basis of the definition of Caputo, the system (11) is simulated by MATLAB, and the attractor phase diagram of system (11) is obtained as shown in Figure 5. It can be seen that the attractor of the system presents obvious chaotic characteristics. Comparing Figure 5 with Figure 1, the interval and shape of the attractor are found to be different.

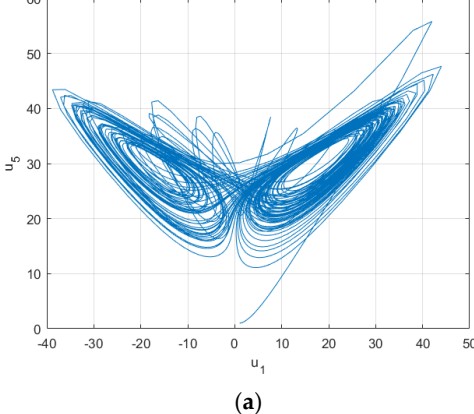

(a)

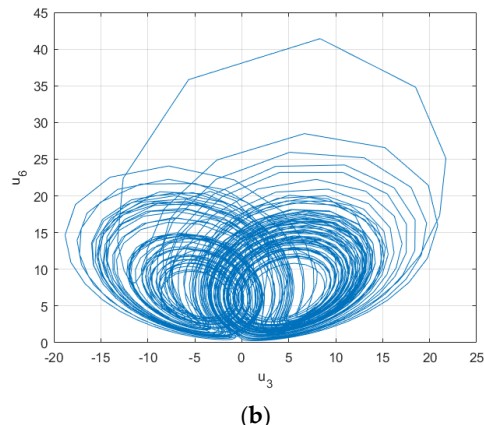

(b)

**Figure 5.** Attractor phase diagram of system (11). (**a**) $u_1 - u_5$ phase diagram of system (11); (**b**) $u_3 - u_6$ phase diagram of system (11).

### 4.4. 0-1 Test

Gottwald and Melbourne proposed a reliable and effective binary test method to test whether the system is chaotic, which is called the "0-1 test" [28]. The basic idea is to establish a stochastic dynamic process for data and then study how the scale of the stochastic process changes with time. Next, we used this method to test and analyze the chaotic characteristics of system (11), as shown in Figure 6.

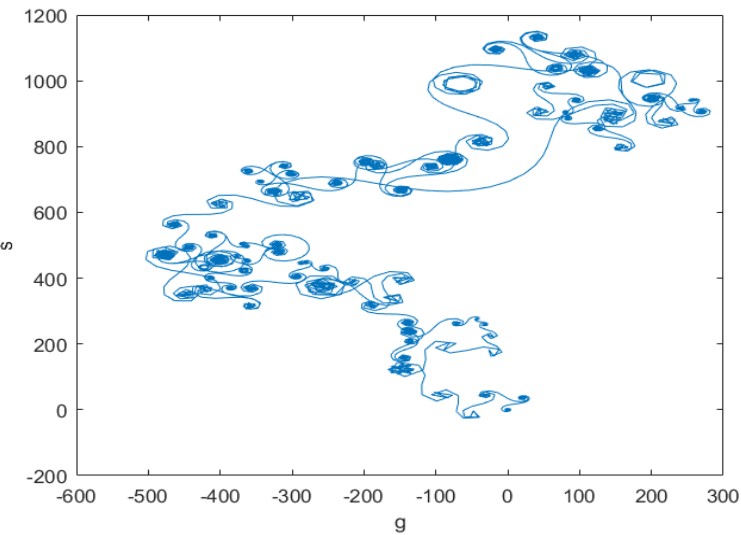

**Figure 6.** The "0-1"test.

It can be seen from Figure 6 that the new hyperchaotic complex system shows obvious unbounded motion, similar to Brownian motion. Therefore, it is chaotic.

### 4.5. Order α Impact on System Status

In this section, we all took $u_1 - u_5$ to observe the evolution of the attractor of the system (11).

When $\alpha < 0.82$, the system is in a divergent state.

When $\alpha \in (0.82, 0.94)$, the system converges to a stable point.

When $\alpha = 0.95$, the system changes from stable point to chaotic state, and at this time, it presents obvious butterfly attractor shape.

When $\alpha = 0.99$, the system is still in chaos, but under the same number of cycles, the shape of the attractor is fuller.

When $\alpha = 1.01$, the order of the system is a fractional order greater than 1. At this time, the system is still in a chaotic state, but the attractor forms are different and sparse.

When $\alpha = 1.03$, the system is in a chaotic state, and the attractor shape becomes fuller with the increase in order.

When $\alpha = 1.05$, the system is in period doubling limit cycle state.

When $\alpha > 1.07$, system divergence occurs.

The detailed evolution process is shown in Figure 7. Through observation, it is found that the fractional complex hyperchaotic system shows a more complex system evolution process with the change of order, such as the position and shape of the attractor having changed to some extent.

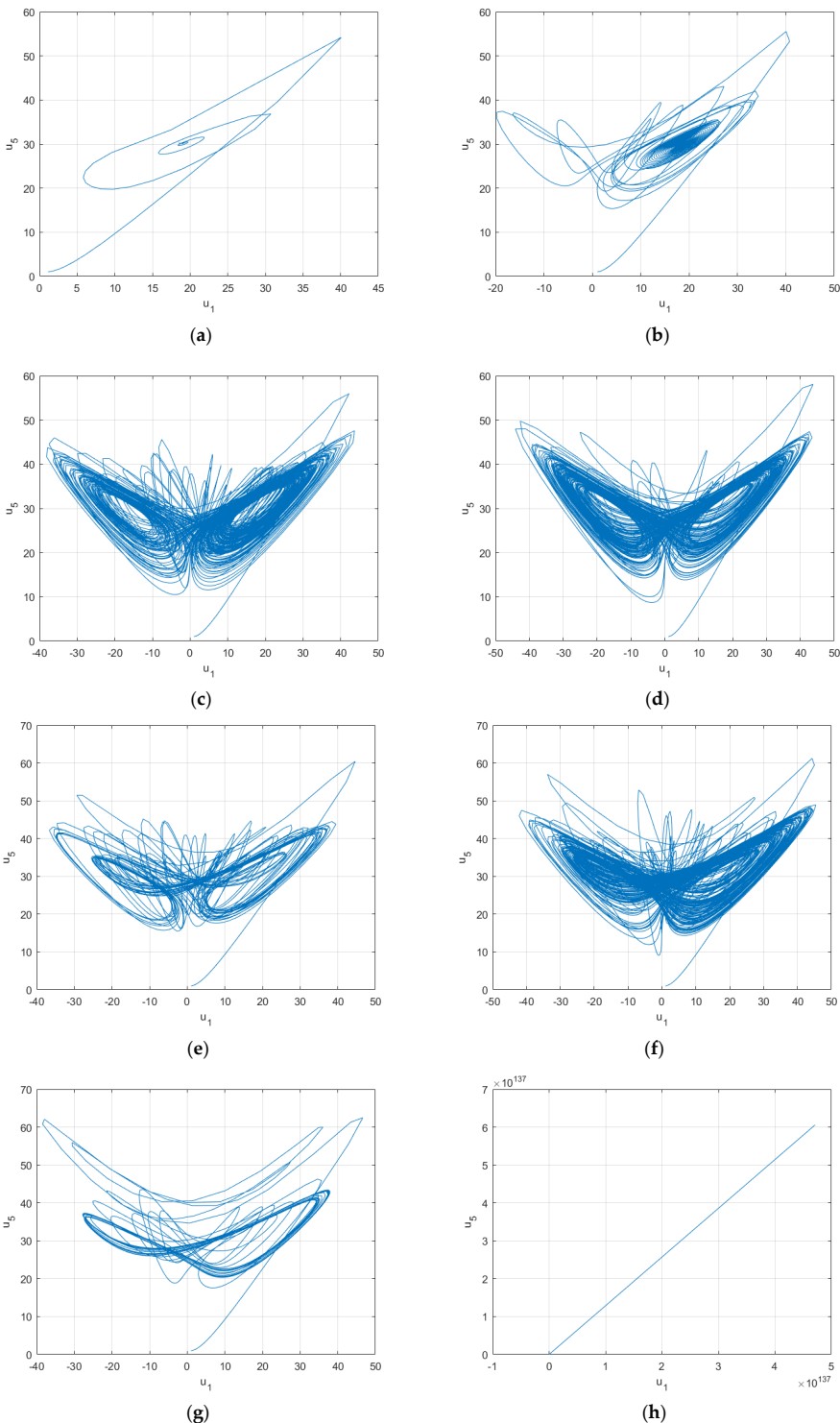

**Figure 7.** The evolution process of system (11) with order. (**a**) $\alpha_{l(l=1,2,3,4)} = 0.82$; (**b**) $\alpha_{l(l=1,2,3,4)} = 0.94$; (**c**) $\alpha_{l(l=1,2,3,4)} = 0.95$; (**d**) $\alpha_{l(l=1,2,3,4)} = 0.99$; (**e**) $\alpha_{l(l=1,2,3,4)} = 1.01$; (**f**) $\alpha_{l(l=1,2,3,4)} = 1.03$; (**g**) $\alpha_{l(l=1,2,3,4)} = 1.05$; (**h**) $\alpha_{l(l=1,2,3,4)} = 1.07$.

## 4.6. Influence of Parameter Change on System Attractor

It can be seen from Section 4.4 that when order $\alpha = 0.95$, the system attractor presents an obvious chaotic attractor form. In this section, we selected order $\alpha = 0.95$ and changed the values of system parameters $a$, $b$, $c$, and $d$ individually to observe the influence of parameters on the system state and system attractor. In this section, we all took the phase diagram of $u_1 - u_5$ to observe the evolution of the attractor of the system.

### 4.6.1. Parameter *a* Change

We kept parameters *b*, *c*, and *d* unchanged; changed the value of parameter *a*; and selected the same initial value as the integer order to observe the evolution process of the system with parameter *a*, as shown in Figure 8. It can be seen that the system entered the chaotic state from the limit cycle state and continued to evolve in the chaotic state. The attractor changed from sparse to full and then to sparse, and then returned to the limit cycle state and finally diverged.

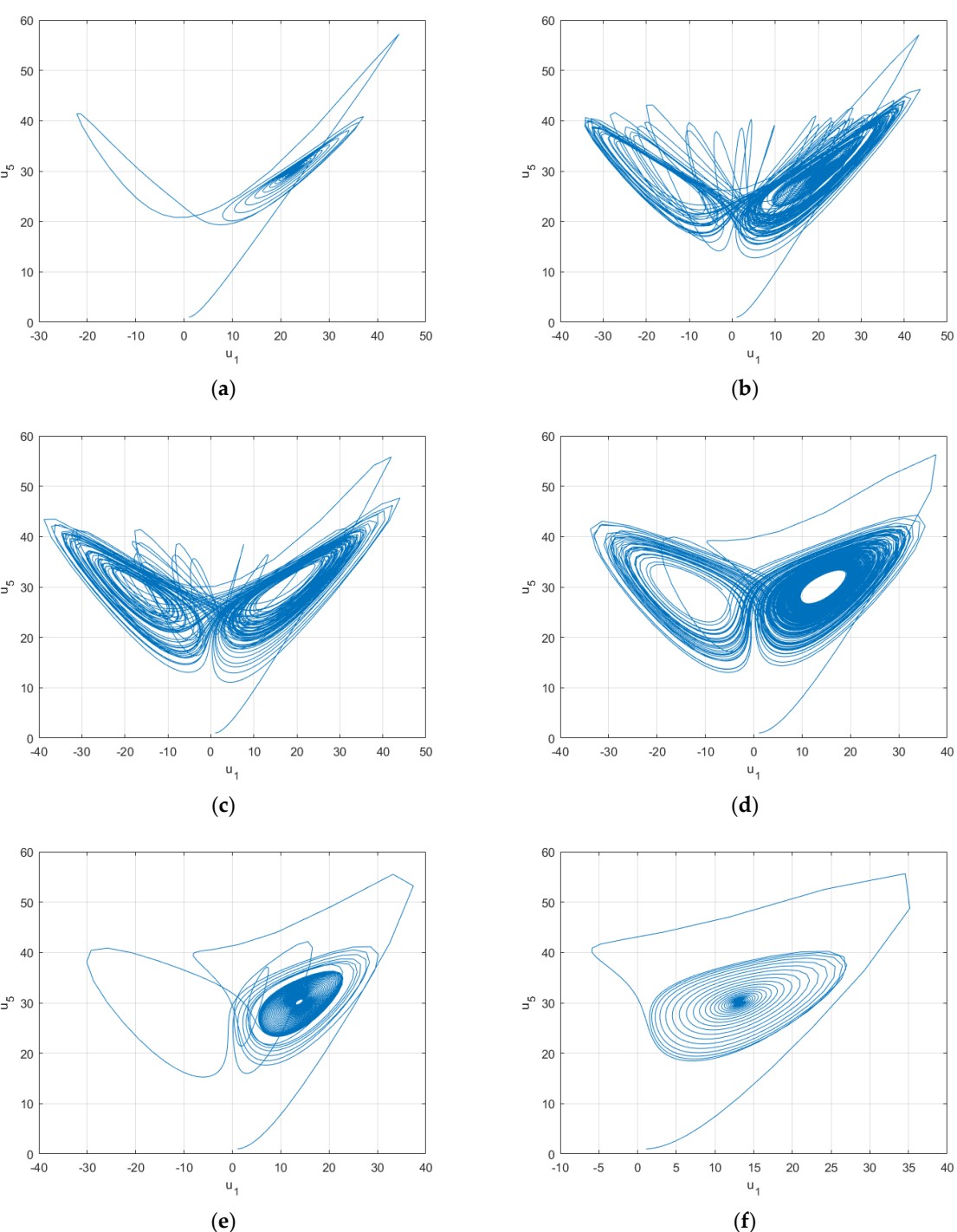

**Figure 8.** The evolution process of system (11) with parameter *a*. (**a**) *a* = 6, limit cycle; (**b**) *a* = 8, chaos; (**c**) *a* = 10, chaos; (**d**) *a* = 20, chaos; (**e**) *a* = 23, chaos; (**f**) *a* = 28, limit cycle.

### 4.6.2. Parameter *b* Change

We kept the parameters *a*, *c*, and *d* unchanged; changed the value of parameter *b*; and selected the same initial value as the integer order to observe the evolution process of the system with the change of parameter *b*, as shown in Figure 9. It can be seen that the system entered the chaotic state from the limit cycle state and continued to evolve in the chaotic state. The attractor changed from sparse to full, then returned to the limit cycle state and finally diverged.

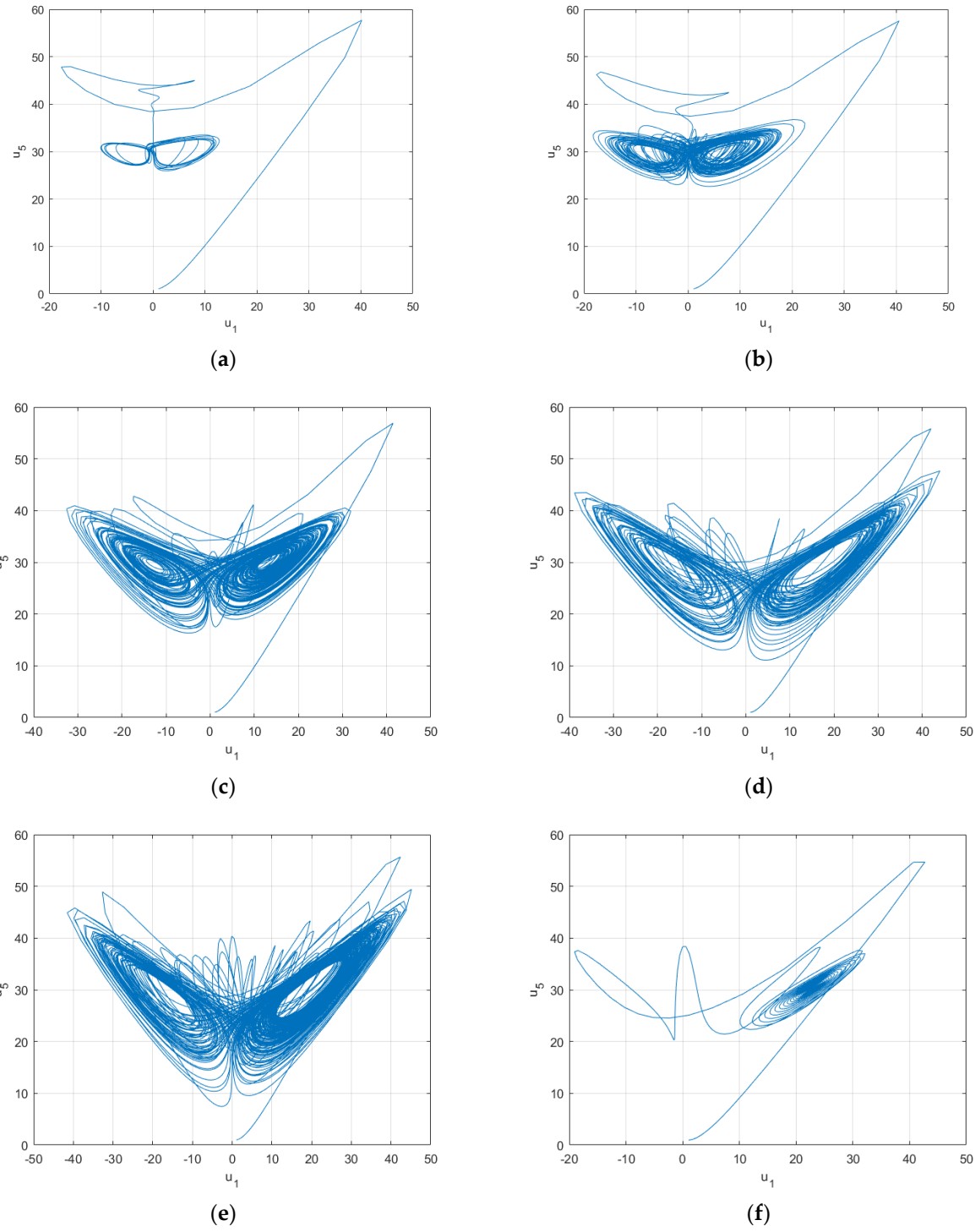

**Figure 9.** The evolution process of system (11) with parameter *b*. (**a**) *b* = 0.2, limit cycle; (**b**) *b* = 0.5, chaos; (**c**) *b* = 1.5, chaos; (**d**) *b* = 8/3, chaos; (**e**) *b* = 3, chaos; (**f**) *b* = 4, limit cycle.

### 4.6.3. Parameter *c* Change

We kept parameters *a*, *b*, and *d* unchanged; changed the value of parameter *c*; and selected the same initial value as the integer order to observe the evolution process of the system with the change of parameter *c*, as shown in Figure 10. It can be seen that the system entered the chaotic state from the limit cycle state, evolved continuously in the chaotic state, and finally returned to the limit cycle state and diverged.

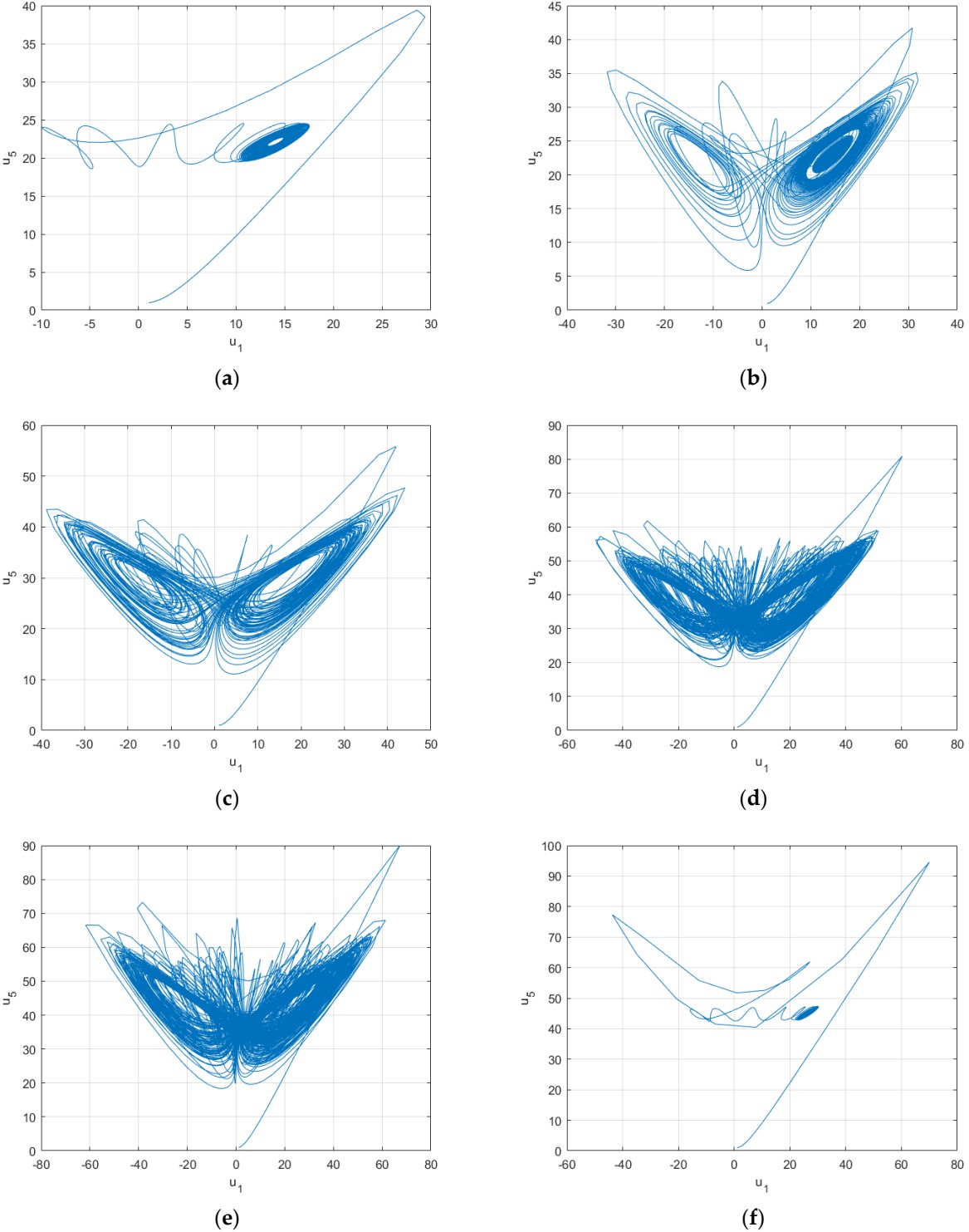

**Figure 10.** The evolution process of system (11) with parameter *c*. (**a**) *c* = 22, limit cycle; (**b**) *c* = 23, chaos; (**c**) *c* = 30, chaos; (**d**) *c* = 40, chaos; (**e**) *c* = 44, chaos; (**f**) *c* = 44.8, limit cycle.

### 4.6.4. Parameter *d* Change

We kept parameters *a*, *b*, and *c* unchanged; changed the value of parameter *d*; and selected the same initial value as the integer order to observe the evolution process of the system with parameter *d*, as shown in Figure 11. It can be seen that the system entered the chaotic state from the limit cycle state and evolved continuously in the chaotic state. It can be seen that when the system was in the chaotic state, compared with parameters *a*, *b*, *c*, parameter *d* had the largest value range, and finally the system returned to the limit cycle state and finally diverged.

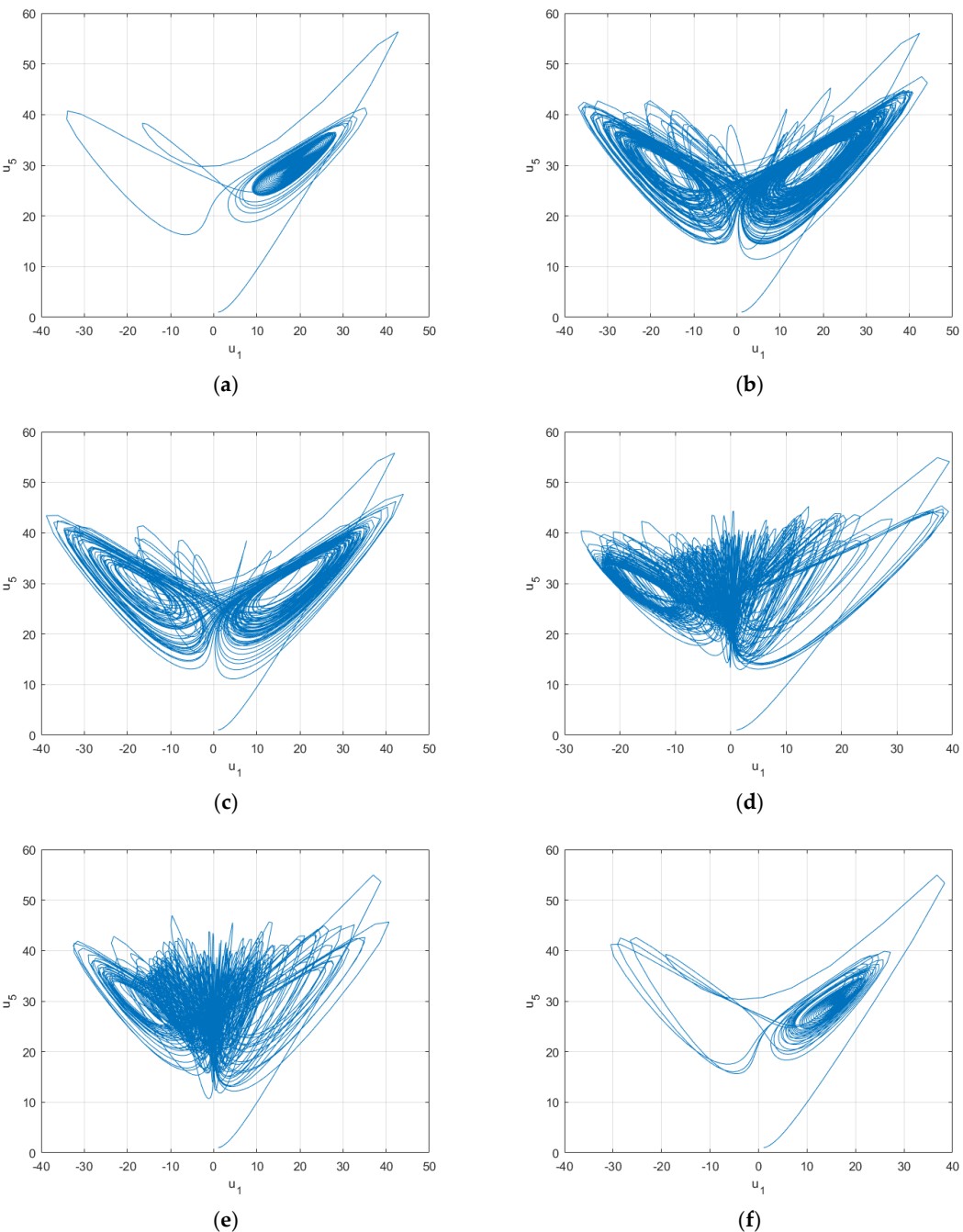

**Figure 11.** The evolution process of system (11) with parameter *d*. (**a**) *d* = 6, limit cycle; (**b**) *d* = 9, chaos; (**c**) *d* = 10, chaos; (**d**) *d* = 60, chaos; (**e**) *d* = 100, chaos; (**f**) *d* = 151.6, limit cycle.

It was found that the fractional hyperchaotic complex system showed a more complex system evolution process with the change of parameters, such as the position and shape of the attractor having changed to some extent.

## 5. Conclusions

In this paper, aiming at the harmful hyperchaotic complex behavior in industrial process, a tracking controller was designed for the hyperchaotic complex system so that the three state variables of the hyperchaotic complex system can track the controller of sine function, constant 5, and complex Lorenz chaotic system individually, and its stability was proven to realize the tracking control of the system. In order to make the intentional chaotic behavior in the industrial process more complex, we extended the hyperchaotic complex system to fractional order. The effects of initial value, order, and parameter changes on the fractional hyperchaotic complex system were discussed and studied. When the order and parameters changed, the detailed evolution process of the system state was given. It was found that there were no coexistence attractors and parameter attractors in the system.

In this paper, the application of the hyperchaotic complex system was studied. For the harmful chaotic system, the controller was designed to convert it into the desired system; for the beneficial chaotic system, this paper extended it to fractional order, which made its chaotic behavior more complex.

There are several prospects for the study of chaos theory: (1) research on the physical background of the chaotic system, or it can show more abundant dynamic behavior; (2) chaos theory can be applied to some complex systems, such as weather forecasting and industrial processes; (3) on the basis of chaotic systems, new chaotic cryptographic algorithms or chaotic neural networks can be formed and applied in various fields.

**Author Contributions:** Conceptualization, F.L., Z.L. and F.Z.; methodology, Z.L. and L.L.; software, F.L., Z.L. and S.Z.; validation, F.L., L.L. and Z.L.; formal analysis, F.L. and L.L.; investigation, F.L. and F.Z.; resources, F.L. and F.Z.; data curation, F.L. and Z.L.; writing—original draft preparation, F.L., L.L. and Z.L.; writing—review and editing, F.L., Z.L. and F.Z.; visualization, F.L., L.L., Z.L. and S.Z.; supervision, F.Z.; project administration, F.Z.; funding acquisition, F.L., Z.L. and F.Z. All authors have read and agreed to the published version of the manuscript.

**Funding:** This work is in part supported by the Shandong Provincial Natural Science Fund (2022HWY Q-081), the National Science Foundation of China Key Project (U21A20488), Academic Promotion Project of Shandong First Medical University, the International Collaborative Research Project of Qilu University of Technology (QLUTGJHZ2018020), and major scientific and technological innovation projects of Shandong Province under grant number (2019JZZY010731 and 2020CXGC010901).

**Institutional Review Board Statement:** Not applicable.

**Informed Consent Statement:** Not applicable.

**Data Availability Statement:** Not applicable.

**Conflicts of Interest:** The authors declare no conflict of interest.

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
