# Peer review of "Tracking Control of a Hyperchaotic Complex System and Its Fractional-Order Generalization"

_processes, doi:10.3390/pr10071244_

Round 1

Reviewer 1 Report

Report on the paper:

Tracking control of a hyperchaotic complex system and its fractional-order generalization

By Feng Liang, Lu Lu, Zhengfeng Li, and Shuaihu Zhang submitted to Processes

This paper extends the hyperchaotic complex system to the fractional-order as the fractional-order has more complex dynamic characteristics.

In my opinion, the proof of the results seems to be corrected. Moreover, in my opinion the result is new and interesting. Accordingly, I recommend it for publication in Processes, after some minor corrections I have indicated below.

1.     I suggest to improve the abstract of the paper.

2.     There are a some typos in the paper that should be corrected by the authors.

3.     I suggest to add the following: https:// doi.org/10.3390/math9212781

4.     What are future recommendations?

Reviewer 2 Report

See attached file
